# Assessment of risk factors for virological nonsuppression following switch to dolutegravir and lamivudine, or bictegravir, emtricitabine, and tenofovir alafenamide fumarate in a real-world cohort of treatment-experienced adults living with HIV

Shu-Yuan Lee[1], Yi-Chun Lin[1], Cheng-Pin Chen[1,2], Shu-Hsing Cheng[1,3], Shu-Ying Chang[4], Shin-Yen Ku[4], Chien-Yu Cheng [1,5]*

1 Department of Infectious Diseases, Taoyuan General Hospital, Ministry of Health and Welfare, Taoyuan City, Taiwan, 2 School of Clinical Medicine, National Yang-Ming Chiao Tung University, Taipei City, Taiwan, 3 School of Public Health, Taipei Medical University, Taipei City, Taiwan, 4 Department of Nursing, Taoyuan General Hospital, Ministry of Health and Welfare, Taoyuan City, Taiwan, 5 Institute of Public Health, School of Medicine, National Yang-Ming Chiao Tung University, Taipei City, Taiwan

* s841060@yahoo.com

## Abstract

Conflicting data exists regarding the baseline determinants of virological nonsuppression outcomes in treatment-experienced people living with human immunodeficiency virus (PWH) switching to antiretroviral treatment (ART) with bictegravir/emtricitabine/tenofovir alafenamide fumarate (BIC/FTC/TAF) or dolutegravir/lamivudine (DTG/3TC) in Asia. This retrospective observational study, conducted at a designated HIV-care hospital from October 2019 to January 2023, aimed to address this gap. We assessed the odds of virological nonsuppression (VNS) at weeks 48 using logistic regression. A total of 988 patients were included, 35 patients (3.5%) with VNS at week 48. Pre-existing primary resistance-associated mutations (RAM) to nucleoside reverse transcriptase inhibitor (NRTI) and non-nucleoside reverse transcriptase inhibitor (NNRTI) were identified in 11.0% (51/465) and 14.4% (67/465), respectively. The identified risk factor was a record of virological failure ≥2 times (AOR 5.32, 95% CI 2.04–13.85), while an HIV viral load <50 copies/mL within the past three months before switch (AOR: 0.27, 95% CI 0.11–0.72) was identified as a protective factor. No cases of acquired drug resistance-associated mutations were detected at week 48. Additionally, DTG/3TC was noninferior to BIC/FTC/TAF in achieving or maintaining HIV RNA levels of <50 copies/mL, within a -10% noninferiority margin in the per-protocol analysis (responder proportion: 98.2% vs. 95.0%, respectively; adjusted treatment difference [95% CI], 3.2% [0.7% to 5.3%]). In conclusion, DTG/3TC and BIC/FTC/TAF demonstrated good effectiveness in a real-world cohort, but frequent virological failure before the switch might impact the benefits of these regimens in the short term of follow-up.

**Data Availability Statement:** All data analyzed or generated during this study are included in the article and its Supporting information file.

**Funding:** The author(s) received no specific funding for this work.

**Competing interests:** The authors have declared that no competing interests exist.

## Introduction

People living with human immunodeficiency virus currently require lifelong antiretroviral therapy, which has significantly improved life expectancy. As a result, HIV infection is now considered one of the communicable chronic diseases. The 2023 guidelines from the US Department of Health and Human Services, the 2023 European AIDS Clinical Society, and the 2022 International Antiviral Society—USA recommend the use of second-generation integrase strand transfer inhibitors (INSTI) as initial antiretroviral regimens. Examples include bictegravir/emtricitabine/tenofovir alafenamide fumarate and dolutegravir/lamivudine for most treatment-naïve individuals with HIV and as a switch option for those who are virologically suppressed [1–3]. These recommendations are based on the demonstrated potent, durable antiviral activity and a high barrier to resistance of these medications. Several studies have demonstrated the suppressive efficacy of a two-drug regimen comprising DTG plus 3TC similarly to a tenofovir disoproxil fumarate (TDF) or tenofovir alafenamide (TAF)-based three-drug regimen [4–6].

HIV RNA levels >50 copies/mL at 6 months after starting therapy and confirmed HIV RNA levels >50 copies/mL with previously undetectable HIV RNA levels are defined virological failure [2]. Although INSTI-based ART has been shown to achieve high rates of virological suppression, the incidence of virological nonsuppression commonly ranges from 10% to 40% [7–10]. This includes both low-level viremia (LLV, HIV RNA levels <50–1000 copies/mL) and virological failure (VF, HIV RNA levels >1000 copies/mL). While the risk factors for virological nonsuppression outcomes are not yet fully understood, they may include the transmission route, HIV RNA level, CD4 lymphocyte count, type and duration of ART administered, and poor adherence leading to the emergence of RAMs to antiretroviral agents.

Most studies on second-generation integrase inhibitors predominantly included treatment-naïve or virologically suppressed patients [4–6], and there has been a scarcity of research analyzing the risk factors for virological nonsuppression and the impact of pre-existing resistance-associated mutations among ART-experienced and viremic individuals living with HIV. Therefore, the primary objective of this study is to investigate the risk factors associated with virological nonsuppression outcome at 48-week after switching to DTG/3TC or BIC/FTC/TAF among treatment-experienced patients who were either virologically suppressed or non-suppressed.

## Materials and methods

This retrospective, observational cohort study was conducted at a designated hospital for HIV care in Taiwan. The study was conducted to include the patients aged 18 years or older with confirmed HIV infection who switched their prior ART regimen to DTG/3TC or BIC/FTC/TAF from October 2019 to January 2022, and individuals were followed up until January 2023. There was no limitation on the duration of prior ART, and the regimens consisted of two NRTIs plus an INSTI, NNRTI, or protease inhibitor (PI) to co-formulated DTG/3TC or BIC/FTC/TAF as maintenance treatment. The exclusion criteria were (1) active tuberculosis, (2) allergic history or high degree of sensitivity to any component or auxiliary material of the research drug. (3) positive HBsAg status (4) those who discontinued ART, were lost to follow-up, or passed away from the final analyses. Subsequently, the individuals were switched to DTG/3TC or BIC/FTC/TAF at the discretion of treating physicians, and all included patients were followed for a 48-week period (per protocol analysis).

The aim was to examine baseline factors associated with virological non-suppression outcomes at week 48 after switching to DTG/3TC or BIC/FTC/TAF. Additionally, changes in body weight, total cholesterol, and triglyceride levels over the course of the study will be

analyzed in included participants. The study received approval from the research ethics committee or institutional review board (registration number: TYGH112029), and patient consent was waived due to the retrospective nature of the study. The data were accessed for research purposes since August 15, 2023, and all data were analyzed without personal sensitive information.

## Data collection and definition

A standardized case record form was utilized to gather information on demographics, sexual preference, body weight, HIV treatment history, HIV regimens prior to the switch, reasons for switching, HBsAg serology, and results of various laboratory investigations. The tests included plasma HIV RNA level, CD4 lymphocyte count, serum creatinine, liver function, lipid profile, and fasting blood glucose or glycated hemoglobin (HbA1c). These tests were conducted every 3–6 months in adherence to the CDC HIV treatment guidelines in Taiwan, and plasma testing conducted within the past three months before inclusion could serve as reports at the time of switch.

Plasma HIV RNA level was measured using the COBAS AmpliPrep TaqMan HIV-1 test version 2.0 (Roche, Mannheim, Germany), with a lower limit of quantitation of fewer than 20 copies/mL. Virological suppression (VS) was defined as HIV RNA levels <50 copies/mL at week 48, with a 12-week window on either side (snapshot). Low-level viremia was defined as at least twice consecutive plasma HIV RNA level measurements of 50–1000 copies/mL, following VS; viral blip, as an isolated plasma HIV RNA level of 50–1000 copies/mL with previous and subsequent HIV RNA levels <50 copies/mL; and virological failure (VF), as a plasma HIV RNA levels exceeding 1,000 copies/mL. Virological non-suppression included LLV, viral blip, and VF.

Genotypic drug resistance testing was performed routinely since October 2012, but it was not compulsory. All HIV-1 major mutations were identified through population sequencing of HIV-1 RNA and RAMs were predicted using the HIVdb program of the Stanford University HIV Drug Resistance Database, aligning with the drug resistance mutation list of the international AIDS Society-USA Consensus Guidelines [11, 12].

## Statistical analysis

The distributions of patients' demographics and baseline characteristics were summarized using descriptive statistics. Categorical variables were compared using either the chi-square test or Fisher's exact test, while continuous variables were assessed using the Mann—Whitney U test.

## Analysis and regression models

A linear regression model was employed to examine the association between changes in weight and lipids, while a logistic regression model was used to assess the risk factors associated with virological non-suppression at week 48. The Hosmer-Lemeshow test and Nagelkerke's R2 were utilized to assess the proportion of VNS that is explained by the independent variables. All $p$ values were two-sided, and statistical significance was set at a $p$ value <0.05. Kaplan–Meier survival analysis was employed to assess and compare the time to treatment discontinuation between the two treatment groups. The analyses were conducted using SPSS software version 24.0 (SPSS Inc., Chicago, IL).

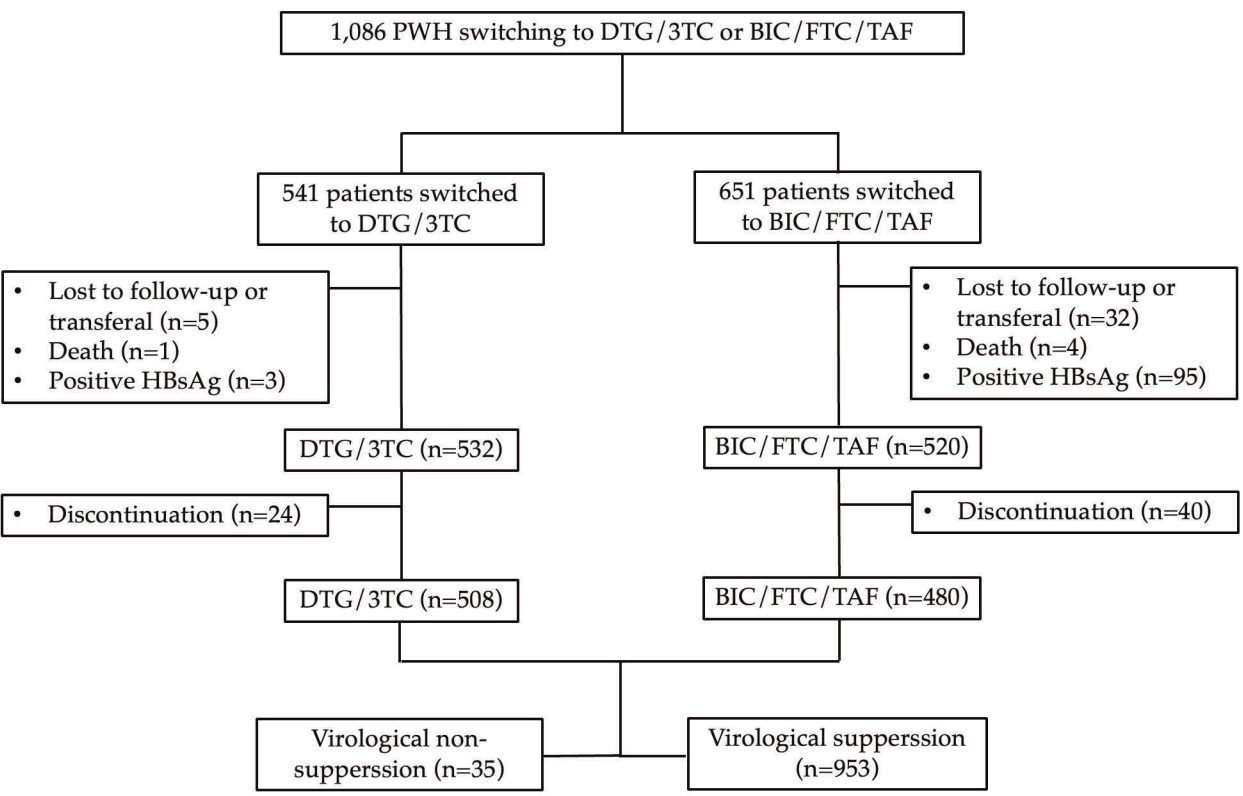

**Fig 1. Flowchart for the inclusion of treatment-experienced patients.** Abbreviation: PWH, people living with HIV; BIC/FTC/TAF, bictegravir/emtricitabine/tenofovir alafenamide; DTG/3TC, dolutegravir/lamivudine.

## Results

### Patient characteristics

Between October 2019 and January 2023, we first screened 1,086 individuals and then included 988 individuals who switched their ART regimen to BIC/FTC/TAF or DTG/3TC, after excluding those who did not meet the inclusion criteria. (Fig 1) Among these, 93.4% were male, 73.0% were men who have sex with men, and 22.6% had a positive HCV antibody. Their median age (IQR) was 38 (32–46) years, and 152 (15.4%) were >50 years old. The median follow-up time since initiating ART was 5.2 (3.1–8.0) years. The mean CD4 counts was 607 cells/μL (434–796), and 34 (3.4%) had CD4 counts <200 cells/μL; 14 (1.4%) had HIV RNA levels >100,000 copies/mL, 77 (7.8%) had HIV RNA levels between 1000 and 100,000 copies/mL, 22 (2.2%) had LLV, and 875(88.6%) had VS at switch.

Moreover, 55 (5.6%) had the record of virological failure more than 2 times before switch. Out of these, 480 (48.6%) individuals switched their ART regimen to BIC/FTC/TAF, while 508 (51.4%) switched to DTG/3TC. VS at weeks 48 was achieved in 953 of 988 (96.5% [95% CI, 95.5%–97.4%]), and baseline characteristics of the individuals are summarized in Table 1.

### Pre-existing primary resistance-associated mutations

Before the switch, 252 (49.6%) PWH had data on RAMs in the DTG/3TC group, and 213 (44.4%) PWH had data in the BIC/FTC/TAF group. Pre-existing primary RAM to any drug

**Table 1. Baseline demographics and clinical characteristics in the study population.**

| | Overall (n = 988) | Virological non-suppression (n = 35) | Virological suppression (n = 953) | p |
|---|---|---|---|---|
| Age, y/o, median (IQR) | 38 (32–46) | 37 (33–40) | 38 (32–46) | 0.046 |
| Male, n (%) | 923 (93.4) | 34 (97.1) | 889 (93.3) | 0.37 |
| HCV Ab positive, n (%) | 213 (22.6) | 7 (20.0) | 206 (22.4) | 0.82 |
| Route of transmission | | | | |
| Men who have sex with men, n (%) | 721 (73.0) | 33 (94.3) | 688 (72.4) | <0.01 |
| Injection drug use, n (%) | 168 (17.0) | 1 (2.9) | 167 (17.3) | 0.02 |
| Heterosexual contact, n (%) | 80 (8.1) | 1 (2.9) | 79 (8.2) | 0.35 |
| Other, n (%) | 19 (1.9) | 0 (0) | 19 (2.0) | 0.39 |
| CD4 counts at switch, cells/uL, median (IQR) | 607 (434–796) | 599 (386–809) | 607 (435–795) | 0.74 |
| CD4 counts >500 cells/uL, n (%) | 614 (62.1) | 22 (62.9) | 592 (621.5) | 0.71 |
| CD4 counts <200 cells/uL, n (%) | 34 (3.4) | 5 (14.3) | 29 (3.0) | <0.01 |
| HIV RNA level at switch | | | | |
| <50 copies/mL, n (%) | 875 (88.6) | 18 (51.4) | 857 (89.0) | <0.01 |
| >100,000 copies/mL, n (%) | 14 (1.4) | 4 (11.4) | 10 (1.0) | <0.01 |
| Low level viremia[1], n (%) | 22 (2.2) | 5 (14.3) | 17 (1.8) | <0.01 |
| Record of virological failure[2], n (%) | | | | |
| Never | 840 (85.0) | 21 (60.0) | 819 (85.01) | <0.01 |
| Once | 148 (15.0) | 14 (40.0) | 134 (13.9) | <0.01 |
| More than 2 times | 55 (5.6) | 11 (31.4) | 44 (4.6) | <0.01 |
| Previous ART regimen | | | | 0.74 |
| Multiple tablet regimen, n (%) | 49 (5.0) | 3 (8.6) | 46 (4.8) | |
| TDF/FTC/EFV, n (%) | 60 (6.1) | 2 (5.7) | 58 (6.0) | |
| BIC/FTC/TAF, n (%) | 11 (1.1) | 0 (0) | 11 (1.1) | |
| TDF/FTC/RPV, n (%) | 14 (1.4) | 1 (2.9) | 13 (1.3) | |
| TAF/FTC/EVG/c, n (%) | 330 (33.4) | 8 (22.9) | 322 (33.4) | |
| DTG/RPV, n (%) | 89 (9.0) | 2 (5.7) | 87 (9.0) | |
| DTG/3TC/ABC, n (%) | 414 (41.9) | 18 (51.4) | 396 (421.1) | |
| TAF/FTC/RPV, n (%) | 21 (2.1) | 2 (5.7) | 19 (2.0) | |
| ART duration before switch, years, median (IQR) | 5.2 (3.1–8.0) | 5.0 (3.3–7.0) | 5.0 (3.1–8.0) | 0.62 |
| ART regimen after switch | | | | 0.02 |
| BIC/FTC/TAF, n (%) | 480 (48.6) | 24 (68.6) | 456 (47.4) | |
| DTG/3TC, n (%) | 508 (51.4) | 11 (31.4) | 497 (51.6) | |

[1]Low-level viremia was characterized by HIV RNA levels between 50 to 1,000 copies/mL occurring more than twice consecutively in the previous year before the switch;

[2]virological failure was defined as a plasma HIV RNA levels exceeding 1,000 copies/mL.

Abbreviation: HCV, hepatitis C virus; ART, anti-retroviral therapy; TDF, Tenofovir fumarate; FTC, emtricitabine; EFV, efavirenz; BIC, bictegravir; TAF, tenofovir alafenamide; RPV, rilpivirine; EVG/c, elvitegravir/cobicistat; DTG, dolutegravir; ABC, abacavir.

class were identified in 24.2% (61/252) of patients in the DTG/3TC group and 31.5% (67/213) in the BIC/FTC/TAF group. The only significant difference in major RAM was M184V/I, with a prevalence of 7.0% in BIC/FTC/TAF compared to 2.4% in DTG/3TC, resulting in a difference of 4.6% (95% CI 1.6%-7.6%, p<0.01). Notably, one patient had K65R+M184V, and 2 patients had Q148H before switch in the DTG/3TC group. In the BIC/FTC/TAF group, three patients had K65R+M184V/I, and one patient had Q148H+G140S before switch (Table 2).

Among PWH switching to DTG/3TC or BIC/FTC/TAF who had pre-existing K65R with or without M184V/I before the switch, all 3 out of 3 (100%) and 3 out of 3 (100%), respectively, had a plasma HIV RNA level of <50 copies/mL at time of switch and successfully maintained

**Table 2.  Primary resistance associated mutations before the switch.**

| | DTG/3TC (n = 252) | BIC/FTC/TAF (n = 213) | | DTG/3TC (n = 252) | BIC/FTC/TAF (n = 213) |
|---|---|---|---|---|---|
| **Primary NRTI-R** | 20 (7.9%) | 31 (14.6%) | **Primary NNRTI-R** | 34 (13.5%) | 33 (15.5%) |
| K65R/E/N | 3 (1.2%) | 3 (1.4%) | L100I | 6 (2.4%) | 3 (1.4%) |
| M184V/I* | 6 (2.4%) | 15 (7.0%) | K101E/P | 4 (1.6%) | 2 (0.9%) |
| L74V | 1 (0.4%) | 2 (0.9%) | K103N/S | 7 (2.8%) | 11 (5.2%) |
| Y115F | 0 | 3 (1.4%) | V106A | 8 (3.2%) | 10 (4.7%) |
| T69NT | 2 (0.8%) | 9 (4.2%) | V108I | 2 (0.8%) | 1 (0.5%) |
| Any TAM[a] | | | E138A/G/K/Q | 7 (2.8%) | 5 (2.3%) |
| D67N | 2 (0.8%) | 5 (2.3%) | V179L | 4 (1.6%) | 5 (2.3%) |
| K70R | 4 (1.6%) | 3 (1.4%) | Y181C | 3 (1.2%) | 4 (1.9%) |
| L210W | 0 | 1 (0.5%) | Y188L | 1 (0.4%) | 5 (2.3%) |
| T215F/Y | 4 (1.6%) | 3 (1.4%) | G190A/Q/S | 3 (1.2%) | 3 (1.4%) |
| K219E/N/Q/R | 1 (0.4%) | 2 (0.9%) | H221Y | 1 (0.4%) | 3 (1.4%) |
| | | | M230I | 1 (0.4%) | 0 |
| **Primary PI-R** | 5 (2.0%) | 1 (0.5%) | **Primary INSTI-R** | 2 (0.8%) | 2 (0.9%) |
| M46I/L | 3 (1.2%) | 1 (0.5%) | T66I | 0 | 1 (0.5%) |
| I54L | 1 (0.4%) | 0 | G140S | 0 | 1 (0.5%) |
| V82A/L | 1 (0.4%) | 0 | Q148H | 2 (0.8%) | 1 (0.5%) |

[a] TAMs are M41L, D67N, K70R, L210W, T215F/Y, and K219E/N/Q/R in RT;

*$p<0.05$

Abbreviation: NRTI, nucleoside reverse transcriptase inhibitor; TAMs, thymidine analog mutations; NNRTI, non-nucleoside reverse transcriptase inhibitor; PI, protease inhibitor; INSTI, integrase strand transfer inhibitor; R, resistance.

<50 copies/mL at week 48. It is worth noting that these 6 patients already achieved VS at baseline. Furthermore, among PWH with pre-existing M184V/I, 15 out of 21 (71.4%) patients had VS at the time of the switch. Among these, 5 out of 6 (83.3%) in the DTG/3TC group and 15 out of 15 (100%) in BIC/FTC/TAF group had an HIV RNA level of <50 copies/mL at week 48. Among the 6 patients with VNS at the time of switch, 6 out of 6 (100%) achieved VS. However, 1 out of 15 (6.7%) patients with VS at switch failed to maintain an HIV RNA level of <50 copies/mL at week 48. For patients with M184V/I, the median duration of previous VS before switch was 3.6 years (IQR: 2.5–5.5). Among individuals with Q148H genotype in the DTG/3TC group, two patients had HIV RNA levels viral loads of 72 and <50 copies/mL at the time of switch, respectively. In the BIC/FTC/TAF group, one patient with Q148H and G140S had an HIV RNA level of 13,700copies/mL at the time of switch. However, all three patients achieved VS by week 48. Moreover, among the 35 patients (3.5%) with VNS at week 48, no one was detected to have acquired drug resistance-associated mutations. Of the 35 patients, only 7 had HIV RNA levels >1000 copies/mL, which is sufficient for detecting RAM, while the remaining 28 had HIV RNA levels between 50 and 1000 copies/mL.

## Risk factors of virological non-suppression at week 48 after switch

As shown in Table 3, VNS were analyzed using a logistic regression model, which showed a Nagelkerke R Square of 0.218 and a Hosmer-Lemeshow test result of 0.582, indicating a moderate fit. After adjusting for baseline age, sex, route of transmission, primary RAM, CD4 cell counts, HIV RNA, low-level viremia, record of virological failure, and ART duration and

**Table 3. The analysis of risk factors of HIV RNA levels ≥ 50 copies/mL at week 48.**

| | Crude Odds Ratio (95% CI) | p | Adjusted Odds Ratio (95% CI) | p |
|---|---|---|---|---|
| Gender, male vs. female | 2.51 (0.34–18.65) | 0.37 | 1.23 (0.10–15.31) | 0.87 |
| Age (years), ≥50 vs. <50 | 0.88 (0.34–2.31) | 0.80 | 0.99 (0.34–2.86) | 0.99 |
| Men who have sex with men, yes vs. no | 4.25 (1.29–13.96) | 0.02 | 4.19 (0.5–35.08) | 0.19 |
| Injection drug use, yes vs. no | 0.28 (0.07–1.16) | 0.08 | 0.49 (0.03–7.03) | 0.60 |
| Positive HCV antibody, yes vs. no | 1.03 (0.46–2.30) | 0.94 | 1.97 (0.73–5.30) | 0.18 |
| Primary NRTI-R, yes vs. no | 3.19 (1.18–8.58) | 0.02 | 2.48 (0.74–8.30) | 0.14 |
| Primary NNRTI-R, yes vs. no | 1.77 (0.61–5.16) | 0.30 | 0.82 (0.22–3.14) | 0.78 |
| CD4 count (cells/μL), <200 vs. ≥200 | 5.15 (1.87–14.20) | <0.01 | 1.39 (0.39–4.95) | 0.61 |
| HIV RNA level (copies/mL), <50 vs. ≥50 | 0.13 (0.06–0.25) | <0.01 | 0.27 (0.11–0.72) | <0.01 |
| Low level viremia, yes vs. no | 8.90 (3.09–26.67) | <0.01 | 2.26 (0.60–8.49) | 0.23 |
| Virological failure ≥2 times, yes vs. no | 9.11 (4.21–19.70) | <0.01 | 5.32 (2.04–13.85) | <0.01 |
| ART duration (years), ≥3 vs. <3 | 0.53 (0.26–1.06) | 0.07 | 0.57 (0.26–1.23) | 0.15 |
| ART regimen, DTG/3TC vs. BIC/FTC/TAF | 0.46 (0.23–0.92) | 0.03 | 0.77 (0.35–1.70) | 0.51 |

Abbreviation: NRTI, nucleoside reverse transcriptase inhibitor; NNRTI, non-nucleoside reverse transcriptase inhibitor; R, resistance; ART, anti-retroviral therapy.

regimens. The following factors before the switch were considered significantly: a record of virological failure ≥ 2 times (AOR 5.32, 95% CI 2.04–13.85), while an HIV viral load <50 copies/mL within the past three months before switch (AOR: 0.27, 95% CI 0.11–0.72) was identified as a protective factor.

At week 48, DTG/3TC was noninferior to BIC/FTC/TAF in achieving HIV RNA <50 copies/mL (-10% noninferiority margin) in the per-protocol analysis (proportion of responders, 98.2% vs 95.0%, respectively; adjusted treatment difference = 3.2%, 95%CI, 0.7% - 5.3%]). Moreover, the incidence rate of developing LLV was 3.5 per 100 person-years of follow-up (PYFU) in the BIC/FTC/TAF group and 2.2 per 100 PYFU in the DTG/3TC group [incidence rate ratio (IRR) = 1.64, 95%CI, 0.77–3.49, $p$ = 0.09].

## Immunological response and clinical outcomes

The mean (standard deviation [SD]) change from baseline to week 48 in CD4 lymphocyte cell count was 63 cells/μL (± 221) in the DTG/3TC group and 51 cells/μL (± 209) in the BIC/FTC/TAF group ($p$ = 0.39). Fig 2 summarizes the changes in body weight and lipid profiles of patients after 48 weeks of switching to DTG/3TC and BIC/FTC/TAF. The absolute mean (SD) weight at baseline versus week 48 was 72.4 (± 13.7) vs. 73.4 (± 13.6) kg in the DTG/3TC group (n = 209) and 70.3 (± 11.9) vs. 72.3 (± 12.5) kg in the BIC/FTC/TAF group (n = 171), respectively. In the BIC/FTC/TAF group, body weight increased by 2.0 kg, and in the DTG/3TC group, body weight increased by 1.1 kg (adjusted difference, 0.9 kg; 95% CI, 0.4–1.4; $p$ = 0.053).

The absolute mean total cholesterol (CHOL), high-density lipoprotein (HDL), low-density lipoprotein (LDL) and mean triglyceride (TG) at baseline were 182 (± 33), 48 (± 13), 121 (± 31) and 142 (± 109) mg/dL in the DTG/3TC group, and 179 (± 34), 48 (± 13), 115 (± 29) and 150 (± 100) mg/dL in the BIC/FTC/TAF group, respectively. The absolute mean CHOL/HDL ratio at baseline was 4.0 (± 1.1) in the DTG/3TC group (n = 376), and 3.9 (± 1) in the BIC/FTC/TAF group (n = 270). Blood lipid parameters did not differ substantially from baseline to week 48, including total cholesterol, high-density lipoprotein, low-density lipoprotein, and triglyceride.

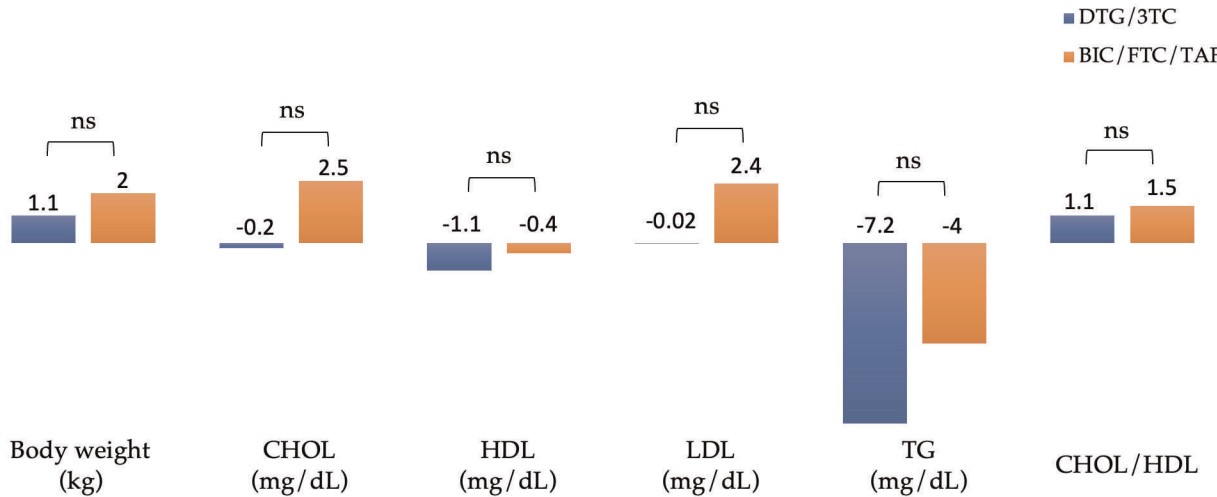

**Fig 2. Changes in body weight and blood lipid from baseline to week 48.** Footnote: The mean difference of body weight (n = 380), CHOL (n = 705), HDL (n = 683), LDL (n = 685), TG (n = 701) and CHOL/HDL (n = 646) between groups were analyzed in the PP-E analysis. Abbreviation: CHOL, total cholesterol; HDL, high density lipoprotein; LDL, low density lipoprotein; TG, triglyceride; PP-E, per-protocol exposed *p <0.05; ns, not significant.

### Adverse events and causes leading to discontinuation

Through week 48, the proportion of patients reporting drug-related adverse events or causes leading to discontinuation was 24 (4.5%) in the DTG/3TC group and 40 (7.7%) in the BIC/FTC/TAF group. The most common adverse events (AEs) in the DTG/3TC group were pruritus (0.9%), insomnia (0.8%), headache/dizziness (0.4%), increased weight (0.4%), and renal function deterioration (0.4%). The most common AEs leading to discontinuation in the BIC/FTC/TAF group were increased weight or obesity (1.5%), insomnia (0.8%), nausea/diarrhea (0.8%) and low-level viremia (0.8%). Treatment discontinuation rates did not significantly differ between the two regimens (Table 4) Fig 3. illustrates the Kaplan–Meier plot displaying the cumulative discontinuation rate, and the hazard ratio for discontinuation of BIC/FTC/TAF was 1.65 (95% CI, 0.99–2.74, *p* = 0.055).

### Discussion

In this real-world study involving a switch to BIC/FTC/TAF or DTG/3TC in 988 ART-experienced PWH, we observed high proportions of suppressed HIV RNA level at week 48. However, the occurrence of HIV RNA levels greater than 50 copies/mL at week 48 after switch was associated with a history of virological failure more than 2 times before the switch. Furthermore, having suppressed HIV RNA level before the switch was associated with a lower risk of HIV RNA levels greater than 50 copies/mL at week 48, even among PWH with pre-existing NRTI resistance.

BIC/FTC/TAF and DTG/3TC are efficacious and well-tolerated INSTI-based regimens for both ART-naïve [5, 13, 14] and ART-experienced people living with HIV [4, 6, 15–19]. They are recommended as first-line ART options in several international HIV treatment. TANGO and SALSA clinical trials revealed that DTG/3TC maintained virological suppression similarly to a TDF or TAF-based three-drug regimen. Our cohort also showed switching to DTG/3TC maintained similar virological suppression as switching to BIC/FTC/TAF at week 48. The proportion of suppressed HIV RNA level was 98% (DTG/3TC) and 98% (BIC/FTC/TAF) in the

**Table 4. The adverse effects and causes leading to discontinuation in the study.**

|  | DTG/3TC (n = 532) | BIC/FTC/TAF (n = 520) |
|---|---|---|
| all events of discontinuation, n (%) | 24 (4.5) | 40 (7.7) |
| renal function deterioration, n (%) | 2 (0.4) | 3 (0.6) |
| numbness, n (%) | 1 (0.2) | 1 (0.2) |
| fatigue, n (%) | 0 | 2 (0.4) |
| headache/dizziness, n (%) | 2 (0.4) | 1 (0.2) |
| anxiety, n (%) | 1 (0.2) | 0 |
| insomnia, n (%) | 4 (0.8) | 4 (0.8) |
| pruritus, n (%) | 5 (0.9) | 2 (0.4) |
| nausea/diarrhea, n (%) | 1 (0.2) | 4 (0.8) |
| blood sugar poor control, n (%) | 0 | 2 (0.4) |
| hyperlipidemia, n (%) | 0 | 3 (0.6) |
| virological failure, n (%) | 1 (0.2) | 2 (0.4) |
| low level viremia, n (%) | 1 (0.2) | 4 (0.8) |
| drug drug interaction, n (%) | 0 | 1 (0.2) |
| increased weight or obesity, n (%) | 2 (0.4) | 8 (1.5) |
| completed LTBI treatment, then shift back to previous ART[1], n (%) | 2 (0.4) | 1 (0.2) |

[1]After completing LTBI treatment, some patients preferred to switch back to their previous ART regimen.

Abbreviation: LTBI, latent tuberculosis infection

subgroup of HIV RNA level <50 copies/mL at the time of switch. Moreover, the interval change from the time of switch in CD4 lymphocyte cell count was not significantly different between the DTG/3TC and BIC/FTC/TAF groups (63 vs. 51 cells/μL, $p$ = 0.39), and the finding was similar to TANGO and SALSA studies.

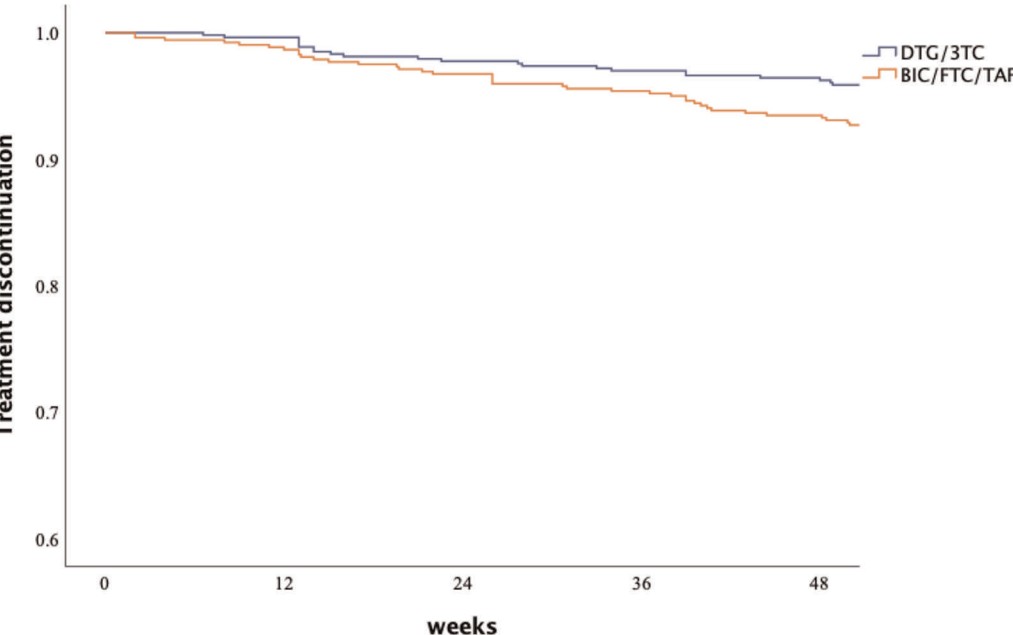

**Fig 3. Kaplan–Meier plot of accumulative treatment discontinuation for DTG/3TC and BIC/FTC/TAF.**

Clinical trials may exhibit the weakness of lacking diversity, with enrolled participants not reflecting the demographics of the intended patient population, thereby limiting the interpretation of published outcomes. On the other hand, retrospective observational cohort studies face the weakness of selection bias. Participants in real-world studies may not be randomly assigned, leading to the potential for a disproportion between groups. This can affect the generalizability of the findings to the broader population [20]. Therefore, we conducted a logistic regression model to identify the risk factors for VNS at week 48. While the TANGO and SALSA studies only enrolled participants limited to VS for more than 6 months, our real-world study included those with VNS at the switch (n = 113, 11.4%). We then proceeded to identify the risk factors in a broader context.

LLV with HIV RNA levels between 50 and 1000 copies/mL could lead to subsequent virological failure (HIV RNA levels >1000 copies/mL) before the era of second-generation integrase strand transfer inhibitor-containing regimens as first-line ART [8, 21, 22]. Chen GJ et al. showed that the risks of developing LLV were similarly low between PWH switched to bictegravir-based (6.2 per 100 person-years) and dolutegravir-based regimens (3.8 per 100 person-years) [IRR = 1.63, 95%CI, 0.90–2.95] [23]. Moreover, the incidences of LLV were 13.2 and 7.0 per 100 person-years of follow-up in the dolutegravir and PI group, respectively [IRR = 1.90, 95% CI, 0.99–3.62] [24]. The results were consistent with our study, which demonstrated that the incidence rate of developing LLV was 3.5 per 100 person-years of follow-up in the BIC/FTC/TAF group and 2.2 per 100 PYFU in the DTG/3TC group [IRR = 1.64, 95%CI, 0.77–3.49].

Some studies confirmed that DTG/3TC and BIC/FTC/TAF were effective, regardless of the existence of the M184V mutation, among suppressed PWH. Andreatta K. et al revealed that at week 48, 98% (561/570) of all BIC/FTC/TAF-treated participants versus 98% (213/217) with pre-existing resistance and 96% (52/54) with archived M184V/I had HIV RNA level <50 copies/mL. No BIC/FTC/TAF-treated participants developed resistance-associated mutations (RAMs) to study drugs [16]. Santoro MM. et al showed that the probability of virological failure and blips in patients switching to DTG/3TC was very low (7.8% and 6.9%) after 3 years of treatment regardless of M184V [25]. Our study found that among PWH switching to DTG/3TC or BIC/FTC/TAF with pre-existing K65R, with or without M184V/I, all seven individuals (100%) had HIV RNA levels less than 50 copies/mL at the time of the switch, and they maintained HIV RNA levels below 50 copies/mL at week 48. However, the effect of a short duration (median of 3.6 years, IQR: 2.5–5.5) of previous virological suppression in patients with M184V/I on DTG/3TC or BIC/FTC/TAF response remains unclear. Therefore, a clinical trial that examines the duration of virological suppression before the switch is warranted.

The mean increase in body weight from the time of switch to week 48 was 2.0 kg in the BIC/FTC/TAF group and 1.1 kg in the DTG/3TC group (adjusted difference, 0.9 kg; 95% CI, 0.4–1.4; $p$ = 0.053). This difference in weight gain between groups may be partly explained by patients in the BIC/FTC/TAF group switching from regimens (TDF/FTC/EFV and TDF/FTC/RPV) known to be associated with weight gain suppression (BIC/FTC/TAF vs DTG/3TC; 11% vs 3%; mean difference, 8%; 95% CI, 5%–11%; $p$ <0.01) [26, 27]. Moreover, overall adverse effects were generally similar between groups, but increased weight or obesity was the leading cause of discontinuation in patients switching to a regimen of BIC/FTC/TAF than those switching to DTG/3TC. Despite the 0.9 kg difference in weight increase, lipid profile was generally unchanged from baseline across both groups.

Our study has several limitations, and the results should be carefully interpreted. First, this was a retrospective study with unbalanced the time of switch characteristics between the two groups, and some of the discontinuations were made by the treating physicians. Therefore, we used a logistic regression model for multivariate analyses to adjust for possible confounding

factors or bias. Second, the small number of VNS at week 48 limited our statistical power when constructing a multivariate model to identify potential risk factors. Third, the included people living with HIV were mainly taking single-tablet regimens before switching to DTG/3TC or BIC/FTC/TAF, and its generalizability was limited to PWH who took multiple-tablet regimens. Fourth, around 50% of all the included PWH had data on resistance-associated mutations before the switch because antiretroviral resistance testing was not routinely available in PWH who initiated or switched ART in Taiwan. Fifth, the small number of patients (6%) with a record of virological failure more than two times before switching can hinder the probability of developing a robust model with sufficient power to achieve accurate predictions. Additionally, with only 35 patients experiencing VNS, the ability to include multiple adjustment variables in the model is constrained, and this restriction can impact the comprehensiveness and predictive accuracy of the model. Sixth, the study was conducted through a per-protocol analysis and not by intention-to-treat. Finally, this was a single-center study that lasted for only 48 weeks, limiting the ability to extrapolate.

## Conclusions

In conclusion, our study suggests that the effectiveness and safety profiles of switching to DTG/3TC and BIC/FTC/TAF were comparable in treatment-experienced adults with HIV, especially among those with VS at the time of switch. However, frequent virological failure before the switch might impact the benefits of these regimens in the short term of follow-up.

## Supporting information

**S1 Raw data.**
(XLSX)

## Author Contributions

**Conceptualization:** Shu-Yuan Lee, Yi-Chun Lin, Cheng-Pin Chen, Shu-Hsing Cheng, Shu-Ying Chang, Shin-Yen Ku, Chien-Yu Cheng.

**Data curation:** Cheng-Pin Chen, Shu-Hsing Cheng, Shu-Ying Chang, Shin-Yen Ku, Chien-Yu Cheng.

**Formal analysis:** Shu-Yuan Lee, Cheng-Pin Chen.

**Investigation:** Shin-Yen Ku, Chien-Yu Cheng.

**Methodology:** Chien-Yu Cheng.

**Project administration:** Chien-Yu Cheng.

**Resources:** Yi-Chun Lin.

**Supervision:** Shu-Hsing Cheng, Chien-Yu Cheng.

**Validation:** Shu-Yuan Lee, Shu-Hsing Cheng, Shu-Ying Chang.

**Writing – original draft:** Shu-Yuan Lee.

**Writing – review & editing:** Cheng-Pin Chen, Shu-Hsing Cheng, Shin-Yen Ku, Chien-Yu Cheng.

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
