## [Decision Letter · Decision Letter 0]

26 Jun 2024

PONE-D-24-08936Assessment of risk factors for virological nonsuppression following switch to dolutegravir and lamivudine, or bictegravir, emtricitabine, and tenofovir alafenamide fumarate in a real-world cohort of treatment-experienced adults living with HIVPLOS ONE

Dear Dr. cheng, Thank you for submitting your manuscript to PLOS ONE. After careful consideration, we feel that it has merit but does not fully meet PLOS ONE’s publication criteria as it currently stands. Therefore, we invite you to submit a revised version of the manuscript that addresses the points raised during the review process.

We look forward to receiving your revised manuscript.

Kind regards,

Dorina Onoya

Academic Editor

PLOS ONE

Journal Requirements:

- https://doi.org/10.1093/jac/dkz347

- http://dx.doi.org/10.1016/j.ijid.2021.02.045

In your revision ensure you cite all your sources (including your own works), and quote or rephrase any duplicated text outside the methods section. Further consideration is dependent on these concerns being addressed.

4. Please include your tables as part of your main manuscript and remove the individual files. Please note that supplementary tables (should remain/ be uploaded) as separate ""supporting information"" files

Reviewers' comments:

Reviewer's Responses to Questions

**Comments to the Author**

1. Is the manuscript technically sound, and do the data support the conclusions?

Reviewer #1: Yes

Reviewer #2: Yes

2. Has the statistical analysis been performed appropriately and rigorously? 

Reviewer #1: No

Reviewer #2: Yes

3. Have the authors made all data underlying the findings in their manuscript fully available?

Reviewer #1: No

Reviewer #2: Yes

4. Is the manuscript presented in an intelligible fashion and written in standard English?

Reviewer #1: Yes

Reviewer #2: Yes

5. Review Comments to the Author

Reviewer #1: The study assessed the real-world effectiveness and safety of switching to either BIC/FTC/TAF or DTG/3TC in a real-world setting, examining both PWH with virological suppression and those without it at the time of switch. The results filled in the gaps left by clinical trials, which typically only included PWH who already had virological suppression. I have some comments as follows.

Major comments:

1. Introduction:

The authors stated that the study aimed to explore the risk factors associated with virological non-suppression following a 48-week period of secondary INSTI-based ART. However, it's important to note that secondary INSTI-based ART includes more than just BIC/FTC/TAF and DTG/3TC.

2. Methods:

The study included PWH who were switched to either BIC/FTC/TAF or DTG/3TC but excluded those who discontinued ART, were lost to follow-up, or passed away from the final analyses. These exclusion criteria should be explicitly stated in the study's methodology. Moreover, PWH who discontinued ART, were lost to follow-up, or passed away could potentially have experience virological non-suppression. Excluding these individuals may introduce bias into the analysis.

3. Results:

(1) In the methodology, the authors indicated that the study was conducted from March 2019 to January 2022, with individuals followed up until January 2023. However, in the results, individuals were screened for inclusion between October 2019 and January 2023.

(2) A logistic regression model was used to clarify the risk factors associated with virological non-suppression. However, the authors included certain variables in the model that might have collinearity, such as a PVL >100,000 cp/mL, a history of virological failure ≥2 times, and PVL <50 cp/mL within the past 3 months before switch. It's crucial to inquire whether the authors assessed collinearity beforehand.

(3) The authors concluded that the effectiveness of DTG/3TC and BIC/FTC/TAF was comparable based on the findings from the multivariable analysis. However, it's still essential to compare effectiveness outcomes, which included virological suppression, low-level viremia, viral blip, and virological failure, between DTG/3TC and BIC/FTC/TAF.

(4) The authors mentioned that no individual was detected to have acquired drug RAMs during follow-up. How many individuals had viremia levels sufficient for detecting RAMs?

(5) Since the tables are not included in the manuscript, I'm having difficulty reviewing the accuracy of the results.

4. Discussion:

The findings that positive HBsAg was related to virological non-suppression should be discussed.

Minor comments:

1. Abstract:

(1) Line 8: A total of 1086 patients were included, 44 patients (4%) with VS at week 48.

-> Please check the number of patients with virological suppression (VS).

(2) In line with the study title, the conclusion should emphasize the impact of various ART regimens rather than focusing on subsequent RAMs.

2. Please standardize the terminology for HIV-1 RNA levels throughout the manuscript. Currently, different terms are used interchangeably, including HIV-1 RNA levels, HIV-VL, plasma viral load, etc.

3. Please ensure that the full names of abbreviations are written out only once, when they first appear in the text.

4. Introduction:

Paragraph 3: Most studies predominantly included treatment-naïve or virologically suppressed patients,

-> Please specify the types of ART studied in the mentioned studies.

5. Methods:

Considering the correlation between positive HBsAg and virological non-suppression, and the decision not to switch PWH coinfected with HBV to DTG/3TC regimen, do you think positive HBsAg should be listed as an exclusion criterion?

6. Results:

(1) Did those 44 patients with detectable viral loads at week 48 continue to use DTG/3TC and BIC/FTC/TAF?

(2) Fig 2. Changes in body weight and blood lipid from baseline to week 48 in the PP-E analysis.

-> What does PP-E analysis stand for?

7. Discussion:

(1) Paragraph 3: "TANGO" should be capitalized.

(2) Paragraph 3: Our real-world study included those with detectable viral loads at the switch (13.2%).

-> However, the results showed that 19 (2%) had PVL ≥100,000 cp/mL and 26 (2%) had LLV at the switch.

(3) Paragraph 4: which demonstrated that the incidence rate of developing LLV was 4.2 per 100 person-years of follow-up in the BIC/FTC/TAF group and 2.3 per 100 PYFU in the DTG/3TC group [incidence rate ratio = 1.83; p = 0.09].

-> The data are not presented in the results part. Additionally, to compare incidences between studies, the authors should briefly list some data from previous studies.

Reviewer #2: PLOS ONE

Assessment of risk factors for virological nonsuppression following switch to dolutegravir and lamivudine, or bictegravir, emtricitabine, and tenofovir alafenamide fumarate in a real-world cohort of treatment-experienced adults living with HIV

Manuscript Number: PONE-D-24-08936

The work presents relevant results from a clinical point of view and adds more information about the change to BIC/FTC/TAF or DTG/3TC from other antiretroviral treatment regimens.

The methodology is correct as well as the statistical analysis. The conclusions are appropriate according to the results presented.

The article is presented in an intelligible fashion and is written in standard English although I am not native and cannot be conclusive in this regard

I believe that the article can be accepted with minimal changes that are suggested below:

Abstract:

VS: Acronyms must be defined. I understand that 44 patients (4%) were virological non suppressors. (VNS)

A switch should be distinguished from undetectability vs. virological failure or treatment suspension and restart with a new regimen.

No significant difference was found between the DTG/3TC and BIC/FTC/TAF regimens (AOR: 1.32, 95% CI 0.59–2.94) (it is not known in the abstract which the OR is favorable towards). That is, using BIC/FTC/TAF multiplies the risk of failure compared to DTG/3TC by 1.32 (with DTG/3TC being the reference) or using DTG/3TC multiplies the risk of failure compared to BIC/FTC/TAF by 1.32

Introduction

The primary objective of this study is to investigate the risk factors associated with virological nonsuppression outcome with secondary INSTI-based ART after a 48-week period. The objective must be defined the population in which the risk factors for virological non-suppression will be analyzed (Patients in Switch). Do it in PICO format. Population (Switch), intervention will change to DTG/3TC (intervention) or BIC/FTC/TAF (comparison) and result (Virological non suppression)

Material and methods.

In exclusion criteria..., were pregnant women excluded? Were patients with HBSag+ not excluded? If not, Were any of them assigned to DTG/3TC?

…with “virological non-suppression” (This terminology should be consistent throughout the manuscript)

In the analysis it must be clear if the end point is through a “snapshot” at week 48 (regardless of what happens up to week 48 or if a confirmed virological failure is performed in a second sample if there is one).

The authors comment: “Virological non-suppression included: LLV, viral blip, and VF”.

In the case of a patient with a VL of 75 copies/ml at w 48 and having previously VL < 50 copies at all times, would it be defined as VF without having analyzed a second VL to know if it is a blip?

Statistic analysis.

Why were parametric tests not carried out instead of non-parametric ones? You had samples of more than 30 subjects in each group. Was a paired data study ever carried out? (McNemar and Wilcoxon? Were they necessary?

In logistic regression models, it would be advisable to add model calibration using the Hosmer-Lemeshow test and Nagelkerke's R2 to assess the proportion of VNS that is explained by the independent variables.

…”Furthermore, 62 (6%) had the record of virological failure more than 2 times before switching”. (There are few patients which can hinder the probability of having a model with sufficient power to achieve accurate predictions (with narrow 95% CI)

Failure was seen in 46 patients, which limits models preferably with no more than 5 adjustment variables...

Just comment in limitations

Results:

Do we know the percentage with RNA+ in those with HCV+ serology?

…Their mean age (SD) was 39.8 (± 9.9)… perhaps better median (p25-p75) years, and 173 (16%) were > 50 years old.

…The mean follow-up time since initiating ART was 6.0 (± 3.7) years… (best median p25-p75).

…The mean CD4 counts was 633 cells/µL (± 307); would be better median and p25-p75,

…19 (2%) had HIV-1 RNA levels ≥100,000 copies/mL, and 26 (2%) had LLV. Did the rest of the patients start with a VL < 50 copies/mL?

Loss of follow up, deaths and discontinuations must be made according to the change to DTG/3TC or BIC/FTC/TAF. A figure would be recommended

Figure 1: the patients who were switched to DTG/3TC or BIC/FTC/TAF should be divided and in the figure express the number who died, discontinued or were lost to follow-up in each group, as well as the number of DTG VNS /3TC or BIC/FTC/TAF

Table 1. Table 1, 2 and 4 are correct.

In Table 3, the reference population is unknown. Especially between DTG/3TC and BIC/FTC/TAF.

Adverse events and causes leading to discontinuation

It would be advisable to create a Kaplan-Meier curve analyzing the discontinuation rate in each arm (DTG/3TC vs TAF/FTC/BIC).

Discussion

Regarding adverse effects, I would comment that surely many of the patients with ABC/3TC/DTG switched to DTG/3TC, which makes it unlikely that clinicians will attribute side effects to the new regimen.

On the contrary, surely most patients from TAF/FTC/EVG/c were transferred to BIC/TAF/FTC, which makes it more likely to attribute toxicity if it exists. Despite this, no greater toxicity was observed in one arm than the other.

Insist again on the way in which the analysis is performed at week 48 (Is it a confirmed virological failure, is it a blip or is it a low level replication...?)

Comment on: HBsAg positive patients should not switch to DTG+3TC due to insufficient HBV control?

Comment in the discussion about this…:

…Furthermore, overall adverse effects were generally similar between groups, but increased weight or obesity was the leading cause of discontinuation in patients switching to a regimen of BIC/FTC/TAF than those switching to DTG/3TC… (Comment: some clinicians attribute increased weights to TAF which may justify more adverse effects or discontinuations of BIC/FTC/TAF without solid data based on clinical trials yet.).

Conclusions

In the conclusions I would say something like:

“In conclusion, our study suggests that the effectiveness and safety profiles of switching to DTG/3TC and BIC/FTC/TAF were comparable in treatment-experienced adults with HIV. The presence of high viral loads, frequent virological failure before the switch and positive HBsAg status might impact the benefits of these regimens in the short term of follow-up”

Bibliografía,

I would include a similar study already evaluated in this journal such as:

Mendoza I, Lázaro A, Espinosa A, Sánchez L, Horta AM, Torralba M. Effectiveness, durability and safety of dolutegravir and lamivudine versus bictegravir, emtricitabine and tenofovir alafenamide in a real-world cohort of HIV-infected adults. PLoS One. 2023 Sep 29;18(9):e0291480. doi: 10.1371/journal.pone.0291480. PMID: 37773939; PMCID: PMC10540944.

6. PLOS authors have the option to publish the peer review history of their article (what does this mean?). If published, this will include your full peer review and any attached files.

Reviewer #1: No

Reviewer #2: **Yes: **Miguel Torralba MD PhD.

---

## [Author Response · Author response to Decision Letter 0]

25 Jul 2024

Dear Editor,

We would like to thank you and the reviewers for giving us the opportunity to improve our manuscript entitled “Assessment of risk factors for virological nonsuppression following switch to dolutegravir and lamivudine, or bictegravir, emtricitabine, and tenofovir alafenamide fumarate in a real-world cohort of treatment-experienced adults living with HIV”. Our point-to-point response to the reviewers' comments and queries are listed below.

Reply: We have revised the manuscript and included the requirements as attachments. 

Reply: We cited these 2 articles in reference 16 and 22, and the resistance mutation were analyzed in these 2 articles and our study, but we believed the outcome is slightly different. 

Reply: We have uploaded all raw data as Supporting Information files.

4. Please include your tables as part of your main manuscript and remove the individual files. Please note that supplementary tables (should remain/ be uploaded) as separate ""supporting information"" files. 

 Reply: We have revised the manuscript and included the requirements as attachments. 

Review Comments to the Author:

Reviewer 1:

Major comments:

1. Introduction:

The authors stated that the study aimed to explore the risk factors associated with virological non-suppression following a 48-week period of secondary INSTI-based ART. However, it's important to note that secondary INSTI-based ART includes more than just BIC/FTC/TAF and DTG/3TC.

Reply: Thank you for your important comment. However, BIC/FTC/TAF and DTG/3TC were prescribed dominantly at our hospital, while the remaining secondary INSTI-based ART prescriptions were less than 1%.

2. Methods: The study included PWH who were switched to either BIC/FTC/TAF or DTG/3TC but excluded those who discontinued ART, were lost to follow-up, or passed away from the final analyses. These exclusion criteria should be explicitly stated in the study's methodology. Moreover, PWH who discontinued ART, were lost to follow-up, or passed away could potentially have experience virological non-suppression. Excluding these individuals may introduce bias into the analysis.

Reply: Your idea is very important. We added a third exclusion criterion for those who discontinued ART, were lost to follow-up, or passed away. Additionally, our study is a per-protocol analysis, and all included patients should take BIC/FTC/TAF or DTG/3TC for a 48-week period.

3. Results:

(1) In the methodology, the authors indicated that the study was conducted from March 2019 to January 2022, with individuals followed up until January 2023. However, in the results, individuals were screened for inclusion between October 2019 and January 2023.

Reply: It’s from October 2019 and January 2023. We revised our Methods in page 4. 

(2) A logistic regression model was used to clarify the risk factors associated with virological non-suppression. However, the authors included certain variables in the model that might have collinearity, such as a PVL >100,000 cp/mL, a history of virological failure ≥2 times, and PVL <50 cp/mL within the past 3 months before switch. It's crucial to inquire whether the authors assessed collinearity beforehand.

Reply: Thank for your valuable idea. We deleted 2 variables of PVL>100,000 copies/ml and HBsAg, because the other reviewer also suggested that positive HBsAg PLWH should be excluded, because only 3 patients with positive HBsAg was switched to DTG/3TC, but 95 patients were switched to BIC/FTC/TAF disproportionally.

(3) The authors concluded that the effectiveness of DTG/3TC and BIC/FTC/TAF was comparable based on the findings from the multivariable analysis. However, it's still essential to compare effectiveness outcomes, which included virological suppression, low-level viremia, viral blip, and virological failure, between DTG/3TC and BIC/FTC/TAF.

Reply: At week 48, DTG/3TC was noninferior to BIC/FTC/TAF in achieving HIV RNA < 50 copies/mL (-10% noninferiority margin) in the per-protocol analysis (proportion of responders, 98.2% vs 95.0%, respectively; adjusted treatment difference [95% CI], 3.2% [0.7& to 5.3%]).

(4) The authors mentioned that no individual was detected to have acquired drug RAMs during follow-up. How many individuals had viremia levels sufficient for detecting RAMs?

Reply: Only 7 individuals had HIV RNA> 1000 copies/mL which is sufficient for detecting RAM, and the rest (n=28) had HIV RNA between 50 to 1000 copies/mL. 

(5) Since the tables are not included in the manuscript, I'm having difficulty reviewing the accuracy of the results.

 Reply: Tables were attached in supplement files, and we revised the tables to our manuscript. 

Minor comments:

1. Abstract:

(1) Line 8: A total of 1086 patients were included, 44 patients (4%) with VS at week 48.

-> Please check the number of patients with virological suppression (VS).

Reply: A total of 988 patients were included, 35 patients (3.5%) with VNS at week 48.

(2) In line with the study title, the conclusion should emphasize the impact of various ART regimens rather than focusing on subsequent RAMs.

Reply: In conclusion, DTG/3TC and BIC/FTC/TAF demonstrated good effectiveness in a real-world cohort, but frequent virological failure before the switch might impact the benefits of these regimens in the short-term follow-up.

2. Please standardize the terminology for HIV-1 RNA levels throughout the manuscript. Currently, different terms are used interchangeably, including HIV-1 RNA levels, HIV-VL, plasma viral load, etc.

Reply: Thank for your advice, and we unify this term as HIV RNA. 

3. Please ensure that the full names of abbreviations are written out only once, when they first appear in the text.

Reply: Sure, we revise the abbreviations written out only once. 

4. Introduction:

Paragraph 3: Most studies predominantly included treatment-naïve or virologically suppressed patients,

-> Please specify the types of ART studied in the mentioned studies.

Reply: We only included treatment-experienced patients who were either virologically suppressed or non-suppressed.

5. Methods:

Considering the correlation between positive HBsAg and virological non-suppression, and the decision not to switch PWH coinfected with HBV to DTG/3TC regimen, do you think positive HBsAg should be listed as an exclusion criterion?

Reply: Thank for your valuable idea. We deleted 2 variables of PVL>100,000 copies/ml and HBsAg, because the other reviewer also suggested that positive HBsAg PLWH should be excluded, because only 3 patients with positive HBsAg was switched to DTG/3TC, but 95 patients were switched to BIC/FTC/TAF disproportionally.

6. Results:

(1) Did those 44 patients with detectable viral loads at week 48 continue to use DTG/3TC and BIC/FTC/TAF?

Reply: Yes, 8 over 11 and 18 over 24 continue to use DTG/3TC or BIC/FTC/TAF, because most of them had blips or LLV, and the physicians and nurses boosted their adherence and encourage them to keep HIV treatment. 

(2) Fig 2. Changes in body weight and blood lipid from baseline to week 48 in the PP-E analysis.

-> What does PP-E analysis stand for?

Reply: Per-Protocol-Exposed analysis

7. Discussion:

(1) Paragraph 3: "TANGO" should be capitalized.

Reply: thank you, we revised. 

(2) Paragraph 3: Our real-world study included those with detectable viral loads at the switch (13.2%).

-> However, the results showed that 19 (2%) had PVL ≥100,000 cp/mL and 26 (2%) had LLV at the switch.

Reply: Exclusion criteria included positive HBsAg, therefore total included patients were revised as 988, and 875 patients with HIV RNA <50 copies/mL at switch; hence, 113 patients (11.3%) with detectable viral loads at switch. 

(3) Paragraph 4: which demonstrated that the incidence rate of developing LLV was 4.2 per 100 person-years of follow-up in the BIC/FTC/TAF group and 2.3 per 100 PYFU in the DTG/3TC group [incidence rate ratio = 1.83; p = 0.09].

-> The data are not presented in the results part. Additionally, to compare incidences between studies, the authors should briefly list some data from previous studies.

Reply: Thank for your advice, and we revised my mauscript in “result” and “discussion”. 

Chen GJ et al. showed that the risks of developing LLV were similarly low between PWH switched to bictegravir-based (6.2 per 100 person-years) and dolutegravir-based regimens (3.8 per 100 person-years) [IRR = 1.63, 95%CI, 0.90-2.95] [22]. Moreover, the incidences of LLV were 13.2 and 7.0 per 100 person-years of follow-up in the dolutegravir and PI group, respectively [IRR = 1.90, 95% CI, 0.99–3.62] [23]. The results were consistent with our study, which demonstrated that the incidence rate of developing LLV was 3.5 per 100 person-years of follow-up in the BIC/FTC/TAF group and 2.2 per 100 PYFU in the DTG/3TC group [IRR = 1.64, 95%CI, 0.77-3.49].

Reviewer 2:

Abstract:

VS: Acronyms must be defined. I understand that 44 patients (4%) were virological non suppressors. (VNS) A switch should be distinguished from undetectability vs. virological failure or treatment suspension and restart with a new regimen.

No significant difference was found between the DTG/3TC and BIC/FTC/TAF regimens (AOR: 1.32, 95% CI 0.59–2.94) (it is not known in the abstract which the OR is favorable towards). That is, using BIC/FTC/TAF multiplies the risk of failure compared to DTG/3TC by 1.32 (with DTG/3TC being the reference) or using DTG/3TC multiplies the risk of failure compared to BIC/FTC/TAF by 1.32

Reply: Positive HBsAg patients were excluded, therefore we analyzed the effectiveness of DTG/3TC and BIC/FTC/TAF. Totally, DTG/3TC was noninferior to BIC/FTC/TAF in achieving HIV RNA < 50 copies/mL (-10% noninferiority margin) in the per-protocol analysis (proportion of responders, 98.2% vs 95.0%, respectively; adjusted treatment difference [95% CI], 3.2% [0.7% to 5.3%]). However, among virologically suppressed patients, the effectiveness of DTG/3TC and BIC/FTC/TAF was 97.7% and 98.0%, respectively; adjusted treatment difference [95% CI], -0.3% [-1.9% to 1.3%]). 

Introduction

The primary objective of this study is to investigate the risk factors associated with virological nonsuppression outcome with secondary INSTI-based ART after a 48-week period. The objective must be defined the population in which the risk factors for virological non-suppression will be analyzed (Patients in Switch). Do it in PICO format. Population (Switch), intervention will change to DTG/3TC (intervention) or BIC/FTC/TAF (comparison) and result (Virological non suppression)

Reply: The primary objective of this study is to investigate the risk factors associated with virological nonsuppression outcome with switching to DTG/3TC or BIC/FTC/TAF after a 48-week period among treatment-experienced patients who were either virologically suppressed or non-suppressed.

Material and methods.

In exclusion criteria..., were pregnant women excluded? Were patients with HBsAg+ not excluded? If not, Were any of them assigned to DTG/3TC?

Reply: Thank for your valuable idea. We deleted 2 variables of PVL>100,000 copies/ml and HBsAg, because the other reviewer also suggested that positive HBsAg PLWH should be excluded, because only 3 patients with positive HBsAg was switched to DTG/3TC, but 95 patients were switched to BIC/FTC/TAF disproportionally. 

…with “virological non-suppression” (This terminology should be consistent throughout the manuscript)

Reply: thank you, we revised.

In the analysis it must be clear if the end point is through a “snapshot” at week 48 (regardless of what happens up to week 48 or if a confirmed virological failure is performed in a second sample if there is one).

Reply: Yes, it’s snapshot, per protocol analysis.

The authors comment: “Virological non-suppression included: LLV, viral blip, and VF”. In the case of a patient with a 

VL of 75 copies/ml at w 48 and having previously VL < 50 copies at all times, would it be defined as VF without having analyzed a second VL to know if it is a blip?

Reply: Yes, it is defined as VNS, because HIV RNA can be measured only 4 times annually according to Taiwan CDC HIV guidelines. 

Statistic analysis.

Why were parametric tests not carried out instead of non-parametric ones? You had samples of more than 30 subjects in each group. Was a paired data study ever carried out? (McNemar and Wilcoxon? Were they necessary?

In logistic regression models, it would be advisable to add model calibration using the Hosmer-Lemeshow test and Nagelkerke's R2 to assess the proportion of VNS that is explained by the independent variables.

Reply: Exactly, we performed non-parametric tests, because the data significantly deviate from normality. A paired data study is not ever carried, so we deleted this sentence. Thank for your advice, and we add the Hosmer-Lemeshow test and Nagelkerke's R2 to assess the proportion of VNS that is explained by the independent variables.

…”Furthermore, 62 (6%) had the record of virological failure more than 2 times before switching”. (There are few patients which can hinder the probability of having a model with sufficient power to achieve accurate predictions (with narrow 95% CI)

Failure was seen in 46 patients, which limits models preferably with no more than 5 adjustment variables...

Just comment in limitations

Reply: Fifth, the small number of patients (6%) with a record of virological failure more than two times before switching can hinder the probability of developing a robust model with sufficient power to achieve accurate predictions. Additionally, with only 35 patients experiencing VNS, the ability to include multiple adjustment variables in the model is constrained, and this restriction can impact the comprehensiveness and predictive accuracy of the model.

Results:

Do we know the percentage with RNA+ in those with HCV+ serology?

Reply: Sorry, we didn’t check the viremic status of patients with positive HCV antibody. 

…Their mean age (SD) was 39.8 (± 9.9)… perhaps better median (p25-p75) years, and 173 (16%) were > 50 years old.

…The mean follow-up time since initiating ART was 6.0 (± 3.7) years… (best median p25-p75).

…The mean CD4 counts was 633 cells/µL (± 307); would be better median and p25-p75,

…19 (2%) had HIV-1 RNA levels ≥100,000 copies/mL, and 26 (2%) had LLV. Did the rest of the patients start with a VL < 50 copies/mL?

Reply: Their median age (IQR) was 38 (32-46) years. The median follow-up time since initiating ART was 5.2 (3.1-8.0) years. The mean CD4 counts was 607 cells/µL (434-796). 

..14 (1%) had HIV RNA levels ≥100,000 copies/mL, 22 (2%) had LLV, and 875(89%) had VS at switch. 

Loss of follow up, deaths and discontinuations must be made according to the change to DTG/3TC or BIC/FTC/TAF. A figure would be recommended

Figure 1: the patients who were switched to DTG/3TC or BIC/FTC/TAF should be divided and in the figure express the number who died, discontinued or were lost to follow-up in each group, as well as the number of DTG VNS /3TC or BIC/FTC/TAF

Reply: Thank for your advice, and we made a change.

Table 1. Table 1, 2 and 4 are correct.

In Table 3, the reference population is unknown. Especially between DTG/3TC and BIC/FTC/TAF.

Reply: We revised Table 3 in the manuscript. 

Adverse events and causes leading to discontinuation

It would be advisable to create a Kaplan-Meier curve analyzing the discontinuation rate in each arm (DTG/3TC vs TAF/FTC/BIC).

Reply: We add Fig 3 as Kaplan-Meier curve analyzing the discontinuation rate in each arm (DTG/3TC vs TAF/FTC/BIC). 

Discussion

Regarding adverse effects, I would comment that surely many of the patients with ABC/3TC/DTG switched to DTG/3TC, which makes it unlikely that clinicians will attribute side effects to the new regimen.

On the contrary, surely most patients from TAF/FTC/EVG/c were transferred to BIC/TAF/FTC, which makes it more li

---

## [Decision Letter · Decision Letter 1]

8 Oct 2024

PONE-D-24-08936R1Assessment of risk factors for virological nonsuppression following switch to dolutegravir and lamivudine, or bictegravir, emtricitabine, and tenofovir alafenamide fumarate in a real-world cohort of treatment-experienced adults living with HIVPLOS ONE

Dear Dr. cheng,

Thank you for submitting your manuscript to PLOS ONE. After careful consideration, we feel that it has merit but does not fully meet PLOS ONE’s publication criteria as it currently stands. Therefore, we invite you to submit a revised version of the manuscript that addresses the points raised during the review process.  We kindly ask you to address the minor comments raised before the manuscript can be accepted. 

We look forward to receiving your revised manuscript.

Kind regards,

Benjamin Chimukangara

Academic Editor

PLOS ONE

Journal Requirements:

Additional Editor Comments:

Thank you for the revised manuscript which provides relevant data for virologic outcomes among treatment experienced individuals switching to DTG and BIC regimens. After additional reviews, some minor comments were raised that require your attention. May I kindly ask you to revise the manuscript in line with comments from the reviewers. To help rapidly assess your revised version, please highlight any changes made in the revised manuscript and include specific page numbers in your responses to the reviewer comments. Thank you.

Reviewers' comments:

Reviewer's Responses to Questions

**Comments to the Author**

1. If the authors have adequately addressed your comments raised in a previous round of review and you feel that this manuscript is now acceptable for publication, you may indicate that here to bypass the “Comments to the Author” section, enter your conflict of interest statement in the “Confidential to Editor” section, and submit your "Accept" recommendation.

Reviewer #1: (No Response)

Reviewer #2: All comments have been addressed

2. Is the manuscript technically sound, and do the data support the conclusions?

Reviewer #1: Yes

Reviewer #2: Yes

3. Has the statistical analysis been performed appropriately and rigorously? 

Reviewer #1: Yes

Reviewer #2: Yes

4. Have the authors made all data underlying the findings in their manuscript fully available?

Reviewer #1: Yes

Reviewer #2: Yes

5. Is the manuscript presented in an intelligible fashion and written in standard English?

Reviewer #1: Yes

Reviewer #2: Yes

6. Review Comments to the Author

Reviewer #1: Some issues I previously commented on have been improved in this revised manuscript. However, there are still some aspects that require revision.

1.Abstract: The identified factors associated with VNS should include both "record of virological failure ≥2 times" and "baseline HIV RNA >50 cp/ml". Please correct the inconsistencies between the AOR and 95% CI in the text and Table 3.

2.Introduction:

Paragraph 3: The authors stated "Most studies predominantly included treatment-naïve or virologically suppressed patients..."

-> Please specify the types of ART regimens used in the studies involving treatment-naïve or virologically suppressed patients and cite the relevant references. This would help improve the manuscript by highlighting the research gap.

3. Results:

(1) "Between October 2019 and January 2023, a total of 988 PWH were screened. We included 1,086 eligible individuals who switched their ART regimen to BIC/FTC/TAF or DTG/3TC. (Fig 1)"

-> This could be improved by stating that you first screened 1,086 individuals and then included 988 after excluding those who did not meet the inclusion criteria.

(2) Please unify the decimal places for data in the Results and Tables.

(3) For the 35 individuals not achieving VS, please describe how many had VF and how many had LLV. In the response letter, the authors mentioned that 7 individuals had HIV RNA >1,000 copies/mL, which was sufficient for detecting RAM, while the remaining 28 had HIV RNA levels between 50 and 1,000 copies/mL. This description should be added to the manuscript.

(4) Table 1: Please check if the following values are correct:

Heterosexual contact 80(7%)

Others 0 (2%)

CD4 counts <200 cells/uL 5 (9%)

(5) "Among PWH switching to DTG/3TC or BIC/FTC/TAF who had pre-existing K65R with or without M184V/I before the switch, all 3 out of 3 (100%) and 4 out of 4 (100%)"

-> In Table 2, a total of 6 PWH had pre-existing K65R, not 7.

(6) "However, 1 out of 158 (65.76%) patients with undetectable viral loadsVS at switch failed to maintain an HIV RNA of <50 copies/mL at week 48. For patients with M184V/I, the median duration of previous undetectable viral loadsVS before switch was 3.6 years (IQR: 2.5-5.5)."

-> These sentences would be clearer if included within the same paragraph related to PWH with pre-existing M184V/I.

(7) Please revise the inconsistency between the AOR and 95% CI in the text and Table 3.

(8) "The most common adverse events (AEs) in the DTG/3TC group were..." "The most common AEs leading to discontinuation in the BIC/FTC/TAF group were..."

-> It seems that the authors collected "AEs and causes leading to discontinuation" rather than just "AEs." Please clarify what was collected in the Methods section.

(9) The most common AEs "and causes" leading to discontinuation in the BIC/FTC/TAF group should also include nausea/diarrhea (0.8%).

(10) Table 4:

- Please check if the following value is correct: drug drug interaction 1 (0.)

- Could the authors provide an explanation for "completed LTBI treatment, then shifted back to previous ART" ?

4. Others:

Throughout the manuscript, "HIV RNA" should be revised to "HIV RNA level."

Reviewer #2: Minor Comments

In the abstract and results, the authors state:

"Furthermore, DTG/3TC was noninferior to BIC/FTC/TAF in achieving HIV RNA < 50 copies/mL (-10% noninferiority margin) in the per-protocol analysis (proportion of responders, 98.2% vs 95.0%, respectively; adjusted treatment difference [95% CI], 3.2% [0.7% to 5.3%])."

However,

In switch studies (such as the one conducted by the authors), it is customary to analyze the primary objective of demonstrating noninferiority by detecting virologic failure. In general, the usual delta used by regulatory agencies is 4%. Given that the majority of subjects in the study start with VL < 50 copies/mL, the delta that should not be exceeded is the 4% virologic failure, not the 10% therapeutic success proposed by the authors. This 10% is typical when attempting to demonstrate noninferiority in studies with naïve patients. In this case (naïve patients), the objective is not failure but therapeutic success.

My recommendation is that if the goal is to demonstrate the noninferiority of DTG/3TC vs BIC/FTC/TAF using BIC/FTC/TAF as the reference regimen, the upper limit of the 95% confidence interval should not exceed the 4% virologic failure against TAF.

As I understand it, 11 out of 508 patients fail with DTG/3TC, and 24 out of 480 patients fail with BIC/FTC/TAF.

According to these results (provided by the authors in Table 1), the risk difference for virologic failure is -2.8% in favor of DTG/3TC (95% CI: -5.159, -0.5103; p= 0.008)). Therefore, not only is the noninferiority of DTG/3TC versus BIC/FTC/TAF demonstrated (since the upper limit of the 95% CI does not reach +4%), but the superiority of DTG/3TC is also demonstrated, as the 95% CI does not cross 0% in the difference in virologic failure rates between the two regimens.

In the statistics section, I would note that the Hosmer-Lemeshow test is used to analyze the calibration of the model, and the Nagelkerke R2 is used to assess the percentage of VNS explained by the independent variables. These results are not shown in the article.

The introduction and materials and methods sections are correct.

In the results section, it states:

"14 (1%) had HIV RNA levels ≥100,000 copies/mL, 22 (2%) had LLV, and 875(89%) had VS at switch." This totals 1+2+89%, which sums to 92%. What about the missing 8%?

Before comparing the patients who failed with those who did not fail in Table 1, I would include a table comparing the DTG/3TC and BIC/FTC/TAF groups.

In the paragraph: "Moreover, the incidence rate of developing LLV was 3.5 per 100 person-years of follow-up (PYFU) in the BIC/FTC/TAF group and 2.2 per 100 PYFU in the DTG/3TC group [incidence rate ratio (IRR) = 1.64, 95% CI, 0.77-3.49].", I would add the "p-value."

In the discussion, I would mention as a limitation that the study was conducted through a per-protocol analysis and not by intention-to-treat.

The conclusion is correct.

The bibliography seems appropriate.

7. PLOS authors have the option to publish the peer review history of their article (what does this mean?). If published, this will include your full peer review and any attached files.

Reviewer #1: No

Reviewer #2: **Yes: **Miguel Torralba

---

## [Author Response · Author response to Decision Letter 1]

26 Oct 2024

Dear Reviewers,

We would like to thank you and the reviewers for giving us the opportunity to improve our manuscript entitled “Assessment of risk factors for virological nonsuppression following switch to dolutegravir and lamivudine, or bictegravir, emtricitabine, and tenofovir alafenamide fumarate in a real-world cohort of treatment-experienced adults living with HIV”. Our point-to-point response to the reviewers' comments and queries are listed below.

Reviewer #1: Some issues I previously commented on have been improved in this revised manuscript. However, there are still some aspects that require revision.

1.Abstract: The identified factors associated with VNS should include both "record of virological failure ≥2 times" and "baseline HIV RNA >50 cp/ml". Please correct the inconsistencies between the AOR and 95% CI in the text and Table 3.

Reply: We resvised the inconsistencies. The identified risk factor was a record of virological failure ≥2 times (AOR 5.32, 95% CI 2.04–13.85), while an HIV viral load <50 copies/mL within the past three months before switch (AOR: 0.27, 95% CI 0.11–0.72) was identified as a protective factor.

2.Introduction:

Paragraph 3: The authors stated "Most studies predominantly included treatment-naïve or virologically suppressed patients..."

-> Please specify the types of ART regimens used in the studies involving treatment-naïve or virologically suppressed patients and cite the relevant references. This would help improve the manuscript by highlighting the research gap. 

Reply: Most studies on second-generation integrase inhibitors predominantly included treatment-naïve or virologically suppressed patients [4-6], and there has been a scarcity of research analyzing the risk factors for virological nonsuppression and the impact of pre-existing resistance-associated mutations among ART-experienced and viremic individuals living with HIV.

3. Results:

(1) "Between October 2019 and January 2023, a total of 988 PWH were screened. We included 1,086 eligible individuals who switched their ART regimen to BIC/FTC/TAF or DTG/3TC. (Fig 1)"

-> This could be improved by stating that you first screened 1,086 individuals and then included 988 after excluding those who did not meet the inclusion criteria.

Reply: Thank for your suggestion, and we revised this statement. Between October 2019 and January 2023, we first screened 1,086 individuals and then included 988 switched their ART regimen to BIC/FTC/TAF or DTG/3TC, after excluding those who did not meet the inclusion criteria.

(2) Please unify the decimal places for data in the Results and Tables.

Reply: Thank for your suggestion, we revised and unify the decimal places for data in Table 1 and Results. 

(3) For the 35 individuals not achieving VS, please describe how many had VF and how many had LLV. In the response letter, the authors mentioned that 7 individuals had HIV RNA >1,000 copies/mL, which was sufficient for detecting RAM, while the remaining 28 had HIV RNA levels between 50 and 1,000 copies/mL. This description should be added to the manuscript.

Reply: We added this sentence to the paragraph “Pre-existing primary resistance-associated mutations” Of the 35 patients, only 7 had HIV RNA > 1000 copies/mL, which is sufficient for detecting RAM, while the remaining 28 had HIV RNA levels between 50 and 1000 copies/mL. 

(4) Table 1: Please check if the following values are correct:

Heterosexual contact 80(7%)

Others 0 (2%)

CD4 counts <200 cells/uL 5 (9%)

Reply: Thank you, we revised the data in Table 1. 

(5) "Among PWH switching to DTG/3TC or BIC/FTC/TAF who had pre-existing K65R with or without M184V/I before the switch, all 3 out of 3 (100%) and 4 out of 4 (100%)"

-> In Table 2, a total of 6 PWH had pre-existing K65R, not 7.

Reply: Thank for your inform, and we revised the data as all 3 out of 3 (100%) and 4 out of 4 (100%), respectively.

(6) "However, 1 out of 158 (65.76%) patients with undetectable viral loadsVS at switch failed to maintain an HIV RNA of <50 copies/mL at week 48. For patients with M184V/I, the median duration of previous undetectable viral loadsVS before switch was 3.6 years (IQR: 2.5-5.5)."

-> These sentences would be clearer if included within the same paragraph related to PWH with pre-existing M184V/I.

Reply: Thank for your suggestion, we moved to the the paragraph related to PWH with pre-existing M184V/I.

(7) Please revise the inconsistency between the AOR and 95% CI in the text and Table 3.

Reply: We resvised the inconsistencies. The identified risk factor was a record of virological failure ≥2 times (AOR 5.32, 95% CI 2.04–13.85), while an HIV viral load <50 copies/mL within the past three months before switch (AOR: 0.27, 95% CI 0.11–0.72) was identified as a protective factor.

(8) "The most common adverse events (AEs) in the DTG/3TC group were..." "The most common AEs leading to discontinuation in the BIC/FTC/TAF group were..."

-> It seems that the authors collected "AEs and causes leading to discontinuation" rather than just "AEs." Please clarify what was collected in the Methods section.

Reply: In the analysis of risk factors for virological non-suppression, we did not include those who discontinued ART. This section primarily describes the side effects and reasons for discontinuation caused by BIC/FTC/TAF and DTG/3TC.

(9) The most common AEs "and causes" leading to discontinuation in the BIC/FTC/TAF group should also include nausea/diarrhea (0.8%).

Reply: Thank for your suggestion, and we revised this sentence. 

(10) Table 4:

- Please check if the following value is correct: drug drug interaction 1 (0.)

Reply: 0.2

- Could the authors provide an explanation for "completed LTBI treatment, then shifted back to previous ART" ?

Reply: Because LTBI treatment contains rifapentine, which can reduce the concentrations of ART, such as TAF/FTC/RPV and RPV/DTG, these ART regimens should be shifted to DTG/3TC or BIC/FTC/TAF for combination treatment. After completing LTBI treatment, some patients may prefer to switch back to their previous ART regimen.

4. Others:

Throughout the manuscript, "HIV RNA" should be revised to "HIV RNA level."

Reply: Thank for your suggestion. We revised this term. 

Reviewer #2: Minor Comments

In the abstract and results, the authors state:

"Furthermore, DTG/3TC was noninferior to BIC/FTC/TAF in achieving HIV RNA < 50 copies/mL (-10% noninferiority margin) in the per-protocol analysis (proportion of responders, 98.2% vs 95.0%, respectively; adjusted treatment difference [95% CI], 3.2% [0.7% to 5.3%])."

 However, in switch studies (such as the one conducted by the authors), it is customary to analyze the primary objective of demonstrating noninferiority by detecting virologic failure. In general, the usual delta used by regulatory agencies is 4%. Given that the majority of subjects in the study start with VL < 50 copies/mL, the delta that should not be exceeded is the 4% virologic failure, not the 10% therapeutic success proposed by the authors. This 10% is typical when attempting to demonstrate noninferiority in studies with naïve patients. In this case (naïve patients), the objective is not failure but therapeutic success. 

 My recommendation is that if the goal is to demonstrate the noninferiority of DTG/3TC vs BIC/FTC/TAF using BIC/FTC/TAF as the reference regimen, the upper limit of the 95% confidence interval should not exceed the 4% virologic failure against TAF. As I understand it, 11 out of 508 patients fail with DTG/3TC, and 24 out of 480 patients fail with BIC/FTC/TAF. According to these results (provided by the authors in Table 1), the risk difference for VNS is -2.8% in favor of DTG/3TC (95% CI: -5.159, -0.5103; p= 0.008)). Therefore, not only is the noninferiority of DTG/3TC versus BIC/FTC/TAF demonstrated (since the upper limit of the 95% CI does not reach +4%), but the superiority of DTG/3TC is also demonstrated, as the 95% CI does not cross 0% in the difference in virologic failure rates between the two regimens.

Reply: In introduction, we mentioned “Most studies predominantly included treatment-naïve or virologically suppressed patients, and there has been a scarcity of research analyzing the risk factors for virological nonsuppression and the impact of pre-existing resistance-associated mutations among ART-experienced and viremic individuals living with HIV. Therefore, the primary objective of this study is to investigate the risk factors associated with virological nonsuppression outcome at 48-week after switching to DTG/3TC or BIC/FTC/TAF among treatment-experienced patients who were either virologically suppressed or non-suppressed. Hence, we think the end point of risk difference between DTG/3TC and BIC/FTC/TAF may not be suitable for our study. 

In the statistics section, I would note that the Hosmer-Lemeshow test is used to analyze the calibration of the model, and the Nagelkerke R2 is used to assess the percentage of VNS explained by the independent variables. These results are not shown in the article.

Reply: As shown in Table 3, VNS were analyzed using a logistic regression model, which showed a Nagelkerke R Square of 0.218 and a Hosmer-Lemeshow test result of 0.582, indicating a moderate fit. 

The introduction and materials and methods sections are correct.

In the results section, it states:

"14 (1%) had HIV RNA levels ≥100,000 copies/mL, 22 (2%) had LLV, and 875(89%) had VS at switch." This totals 1+2+89%, which sums to 92%. What about the missing 8%?

Reply: 14 (1.4%) had HIV RNA levels >100,000 copies/mL, 77 (7.8%) had HIV RNA levels between 1000 and 100,000 copies/mL, 22 (2.2%) had LLV, and 875(88.6%) had VS at switch.

Before comparing the patients who failed with those who did not fail in Table 1, I would include a table comparing the DTG/3TC and BIC/FTC/TAF groups.

Reply: We have provided this table based on your recommendation. However, we mentioned in the introduction that the primary objective of this study is to investigate the risk factors associated with virological nonsuppression outcomes at 48 weeks following a switch to DTG/3TC or BIC/FTC/TAF among treatment-experienced patients, whether virologically suppressed or non-suppressed. Therefore, this table may not be suitable for our study.

In the paragraph: "Moreover, the incidence rate of developing LLV was 3.5 per 100 person-years of follow-up (PYFU) in the BIC/FTC/TAF group and 2.2 per 100 PYFU in the DTG/3TC group [incidence rate ratio (IRR) = 1.64, 95% CI, 0.77-3.49].", I would add the "p-value."

Reply: Thank for your suggestion, p=0.09.

In the discussion, I would mention as a limitation that the study was conducted through a per-protocol analysis and not by intention-to-treat.

Reply: Thank for your suggestion, we add this limitation. Sixth, the study was conducted through a per-protocol analysis and not by intention-to-treat.

The conclusion is correct.

The bibliography seems appropriate.

---

## [Editor Report · Decision Letter 2]

31 Oct 2024

PONE-D-24-08936R2Assessment of risk factors for virological nonsuppression following switch to dolutegravir and lamivudine, or bictegravir, emtricitabine, and tenofovir alafenamide fumarate in a real-world cohort of treatment-experienced adults living with HIVPLOS ONE

Dear Dr. cheng,

Thank you for addressing all the comments raised by the reviewers. Please make these minor revisions before we reach a final decision on the manuscript.

We look forward to receiving your revised manuscript.

Kind regards,

Benjamin Chimukangara

Academic Editor

PLOS ONE

Journal Requirements:

Additional Editor Comments:

**General comments**

Please be consistent with the way the < and > signs are used. For example, in introduction, "HIV RNA levels > 50 copies/mL at 6 months after starting therapy and confirmed HIV RNA levels > 50 copies/mL with previously undetectable HIV RNA levels are defined virological failure [2]." Then in another sentence, "This includes both low-level viremia (LLV, HIV RNA levels <50–1000 copies/mL) and virological failure (VF, HIV RNA levels >1000 copies/mL)." I would suggest deleting the space after the sign as done in the second example, and being consistent throughout the manuscript.

**Results**

I would suggest rephrasing to say, "Between October 2019 and January 2023, we first screened 1,086 individuals and then included 988 individuals who switched their ART regimen to BIC/FTC/TAF or DTG/3TC, after excluding those who did not meet the inclusion criteria.

Please consider revising the sentence, "It's worth noting that these 7 patients already achieved VS at baseline." It is still unclear how the patients were 7 if there were 3 in each group with K65R. Also please use the words "It is.." rather than "It's ..."

For clarity, please include a footnote under table 4 for "completed LTBI treatment, then shift back to previous ART, n (%)", to note that "After completing LTBI treatment, some patients preferred to switch back to their previous ART regimen.

Reviewers' comments:

N/A

---

## [Author Response · Author response to Decision Letter 2]

31 Oct 2024

Dear Editor,

We would like to thank you and the reviewers for giving us the opportunity to improve our manuscript entitled “Assessment of risk factors for virological nonsuppression following switch to dolutegravir and lamivudine, or bictegravir, emtricitabine, and tenofovir alafenamide fumarate in a real-world cohort of treatment-experienced adults living with HIV”. Our point-to-point response to the reviewers' comments and queries are listed below.

General comments

Please be consistent with the way the < and > signs are used. For example, in introduction, "HIV RNA levels > 50 copies/mL at 6 months after starting therapy and confirmed HIV RNA levels > 50 copies/mL with previously undetectable HIV RNA levels are defined virological failure [2]." Then in another sentence, "This includes both low-level viremia (LLV, HIV RNA levels <50–1000 copies/mL) and virological failure (VF, HIV RNA levels >1000 copies/mL)." I would suggest deleting the space after the sign as done in the second example, and being consistent throughout the manuscript.

Reply: Thank for your suggestion, and we have revised and made the document consistent according to your recommendation. 

Results

I would suggest rephrasing to say, "Between October 2019 and January 2023, we first screened 1,086 individuals and then included 988 individuals who switched their ART regimen to BIC/FTC/TAF or DTG/3TC, after excluding those who did not meet the inclusion criteria.

Reply: Thank for your suggestion, and we have revised and made the document consistent according to your recommendation. 

Please consider revising the sentence, "It's worth noting that these 7 patients already achieved VS at baseline." It is still unclear how the patients were 7 if there were 3 in each group with K65R. Also please use the words "It is.." rather than "It's ..."

Reply: It is worth noting that these 6 patients already achieved VS at baseline.

For clarity, please include a footnote under table 4 for "completed LTBI treatment, then shift back to previous ART, n (%)", to note that "After completing LTBI treatment, some patients preferred to switch back to their previous ART regimen.

Reply: Thank for your suggestion, and we have revised and made the document consistent according to your recommendation.

---

## [Editor Report · Decision Letter 3]

5 Nov 2024

Assessment of risk factors for virological nonsuppression following switch to dolutegravir and lamivudine, or bictegravir, emtricitabine, and tenofovir alafenamide fumarate in a real-world cohort of treatment-experienced adults living with HIV

PONE-D-24-08936R3

Dear Dr. Chien-Yu Cheng,

We’re pleased to inform you that your manuscript has been considered scientifically suitable for publication and will be formally accepted for publication once it meets all outstanding technical requirements. Congratulations.

Kind regards,

Benjamin Chimukangara

Academic Editor

PLOS ONE

Additional Editor Comments (optional):

None

Reviewers' comments:

None

---

## [Editor Report · Acceptance letter]

11 Nov 2024

PONE-D-24-08936R3 

PLOS ONE

Dear Dr. Cheng, 

I'm pleased to inform you that your manuscript has been deemed suitable for publication in PLOS ONE. Congratulations! Your manuscript is now being handed over to our production team.

Kind regards, 

on behalf of

Dr. Benjamin Chimukangara 

Academic Editor

PLOS ONE